# The Outcomes of Preterm Infants with Neonatal Respiratory Distress Syndrome Treated by Minimally Invasive Surfactant Therapy and Non-Invasive Ventilation

**DOI:** 10.3390/biomedicines12040838

**Published:** 2024-04-10

**Authors:** Tzyy-Rong Huang, Hsiu-Lin Chen, Shu-Ting Yang, Pin-Chun Su, Hao-Wei Chung

**Affiliations:** 1Respiratory Therapy Team, Department of Internal Medicine, Kaohsiung Medical University Hospital, Kaohsiung 80756, Taiwan; perle30300@gmail.com; 2Department of Pediatrics, Kaohsiung Medical University Hospital, Kaohsiung 80756, Taiwan; staceyyang0906@gmail.com (S.-T.Y.); 1020407@gap.kmu.edu.tw (P.-C.S.); plushaowei@gmail.com (H.-W.C.); 3Department of Respiratory Therapy, College of Medicine, Kaohsiung Medical University, Kaohsiung 80708, Taiwan

**Keywords:** non-invasive ventilation, nasal continuous positive airway pressure, nasal intermittent positive pressure ventilation, minimally invasive surfactant therapy, neonatal respiratory distress syndrome

## Abstract

In recent years, the utilization of minimally invasive surfactant therapy (MIST) and Non-invasive ventilation (NIV) as the primary respiratory assistance has become increasingly prevalent among preterm infants with neonatal respiratory distress syndrome (RDS). This study aims to compare the outcomes between MIST administered with nasal continuous positive airway pressure (NCPAP) versus nasal intermittent positive pressure ventilation (NIPPV), with the objective of exploring the respiratory therapeutic benefits of these two approaches. This retrospective study collected data from the neonatal intensive care unit of Kaohsiung Medical University Hospital spanning from January 2016 to June 2021. Infants were divided into two groups based on the type of NIV utilized. The NCPAP group comprised 32 infants, while the NIPPV group comprised 22 infants. Statistical analysis revealed significant differences: the NIPPV group had a smaller gestational age, lower birth weight, higher proportion of female infants, and earlier initiation of MIST. Additionally, the NIPPV group exhibited higher incidence rates of retinopathy of prematurity, longer respiratory support duration, prolonged hospitalization, and mortality. However, upon adjustment, these differences were not statistically significant. Analysis of venous blood gas and respiratory parameter changes indicated that both the NCPAP and NIPPV groups experienced improvements in oxygenation and ventilation following MIST.

## 1. Introduction

Neonatal respiratory distress syndrome (RDS) is a common condition characterized by lung immaturity and insufficient secretion of surfactant among preterm infants. This leads to lung collapse and incomplete expansion after birth, resulting in symptoms such as nasal flaring, chest retractions, and unstable blood oxygen levels.

Previously, surfactant therapy required intubation with endotracheal tube and mechanical ventilator to treat neonatal RDS. Invasive respiratory treatments have been associated with complications in preterm infants, including bronchopulmonary dysplasia (BPD), neurological damage, and retinopathy of prematurity (ROP), as indicated in numerous studies [1,2,3,4,5,6,7,8]. Consequently, there has been a growing focus on the development and research of less invasive surfactant therapy in recent years.

Less invasive surfactant therapy techniques currently used in clinical practice include InSurE (intubation-surfactant-extubation), LISA (less invasive surfactant administration), MIST (minimally invasive surfactant therapy), and others. In recent systematic reviews, these less invasive surfactant therapy methods have been shown to significantly reduce the risk of BPD development and mortality in preterm infants [9]. They also decrease the likelihood of intubation and complications within 72 h of birth and reduce in-hospital mortality. Additionally, a systematic review and network meta-analysis highlighted that, among various non-invasive respiratory support, such as nasal intermittent positive pressure ventilation (NIPPV), nasal continuous positive airway pressure (NCPAP), and high-flow nasal cannula (HFNC), NIPPV is the optimal choice for reducing the risk of BPD and mortality in preterm infants with RDS [10].

However, there is limited research on the combination of less invasive surfactant therapy techniques and non-invasive respiratory support in preterm infants. Therefore, this study aims to explore the therapeutic benefits of applying MIST in neonates with RDS who also received NIPPV or NCPAP as the primary respiratory support.

The analysis aims to evaluate the respiratory treatment outcomes in preterm infants, including the incidence of invasive respiratory support within 72 h, the dosage of administered surfactant, the risk of complications such as BPD, ROP, and intraventricular hemorrhage (IVH), as well as the total length of hospital stay and mortality rate.

## 2. Materials and Methods

### 2.1. Ethics Statement and Study Overview

This retrospective study underwent review by the institutional review board of Kaohsiung Medical University Hospital (KMUHIRB-SV(II)-20210105, approved on 28 December 2021). Medical records were retrieved from 1 January 2014 to 30 June 2021 at the neonatal intensive care unit, Kaohsiung Medical University Hospital.

The preterm infants meeting inclusion criteria for this study were as follows: (1) Neonates diagnosed with neonatal RDS by physicians. (2) Neonates who received ventilatory support via NCPAP or NIPPV for neonatal RDS. (3) Neonates with neonatal RDS who, as per health insurance regulations, required surfactant therapy. The exclusion criteria included individuals who did not meet the aforementioned inclusion criteria, individuals with severe congenital abnormalities, and individuals requiring respiratory support for any other reasons.

After preterm newborns were born and diagnosed with RDS, neonatologists made decisions based on clinical manifestations regarding the choice of NIPPV or NCPAP. Respiratory therapists assisted in setting up the equipment, evaluating clinical indicators such as breath sounds, respiratory patterns, and vital signs. NIV parameters were adjusted by respiratory therapists after discussion with neonatologists.

This study retrospectively collected the clinical characteristics of newborns, including gestational age, birth weight, gender, mode of delivery, and Apgar scores. Relevant parameters related to NIPPV or NCPAP, the use of methylxanthines, blood gas analysis data (collected as part of routine clinical procedures upon admission and at 24 h after admission), frequency of surfactant therapy, and occurrence of complications were collected.

### 2.2. Non-Invasive Ventilators, Ventilator Accessories, and Surfactant Therapy

#### 2.2.1. Non-Invasive Ventilators

NCPAP: (1) Utilized an oxygen blender connected to an underwater continuous positive airway pressure (CPAP) (infant bubble CPAP nasal prongs set, GaleMed™, Taipei, Taiwan). (2) setting parameters: FiO_2_, PEEP, and mixed total flow.NIPPV: (1) The ventilators used in this study included E360 (Newport™ e360 ventilator, Medtronic, Minneapolis, MN, USA) and NPB840 (Puritan Bennett™ 840 ventilator, Medtronic, Minneapolis, MN, USA). For all enrolled subjects in this study, infants over 1 kg were provided with E360, while those weighing less than 1 kg received NPB840. (2) Ventilator mode: pressure-synchronized intermittent mandatory ventilation with pressure support (PSIMV + PS). (3) Setting parameters: peak inspiratory pressure (PIP)/pressure control ventilation (PCV) level, pressure support ventilation (PSV) level, rate, Ti, FiO_2_, and PEEP.

#### 2.2.2. Ventilator Accessories

A nasal prongs set was utilized for both NCPAP and NIPPV groups (infant bubble CPAP nasal prongs set, GaleMed™, Taipei, Taiwan). The ventilator circuit employed was a disposable dual-heated circuit (Evaqua 2 infant circuit RT268, Fisher&Paykel Healthcare, Auckland, New Zealand).

#### 2.2.3. Surfactant

The exogenous surfactant utilized in this study was Survanta^®^ (AbbVie, North Chicago, IL, USA), extracted from natural bovine lungs and administered at a dosage of 100 mg of phospholipids per kilogram of birth weight (4 mL/kg). The initial dose is recommended within 8 h after birth, with a maximum of four doses within the first 48 h.

### 2.3. Minimally Invasive Surfactant Therapy

#### 2.3.1. Indication

According to the reimbursement regulations of the National Health Insurance in Taiwan, newborn infants diagnosed with RDS are eligible to receive surfactant therapy if, despite utilizing artificial ventilation with an oxygen fraction (FiO_2_) set at 40% or higher, they are unable to maintain an arterial oxygen pressure (PaO_2_) greater than 80 mmHg or an oxygen ratio between arteries and alveoli (O_2_ artery/O_2_ alveoli) less than 0.2. 

In this study, when preterm infants with RDS were utilizing NIPPV or NCPAP, and their oxygenation status met the reimbursement criteria of the National Health Insurance, neonatologists would proceed with MIST.

#### 2.3.2. Procedure

Before administering MIST, nurses assisted with suctioning to ensure airway secretions were cleared. They then helped position the newborn in a sniffing position, while the neonatologist, using a laryngoscope, verified the position of the glottis. Subsequently, an angiocatheter (16G) was passed through the glottis into the airway. Once the catheter was in place, the laryngoscope was then removed, and surfactant was slowly administered through the angiocatheter to the lungs. Throughout the MIST procedure, the newborn continued to receive NIV support. The respiratory therapist closely monitored vital signs and ventilator parameters, ensuring optimal oxygenation during drug administration, and making timely adjustments to the ventilator settings as necessary.

### 2.4. Bronchopulmonary Dysplasia

In recent years, HFNC has become increasingly common in neonatal respiratory treatment. However, it was not included in the consensus reached by NICHD in 2001. Therefore, Jensen et al. [7] proposed new BPD classification criteria in 2019 at NICHD. This study defines BPD based on the retrospective enrollment time and corresponding criteria.

### 2.5. Severity of RDS

According to the extent of lung collapse, chest radiographs can demonstrate four grades: (1) Grade I: Bilateral diffuse ground-glass appearance. (2) Grade II: Bilateral diffuse ground-glass appearance with air bronchogram, with differentiation still possible between lung and heart margins. (3) Grade III: Bilateral diffuse ground-glass appearance with air bronchogram, with one side of lung and heart margins indistinguishable. (4) Grade IV: Bilateral white-out, with complete inability to distinguish between lung and heart margins on both sides.

### 2.6. Statistical Analysis

Personal identifiers were replaced with codes, and file creation was performed using Microsoft^®^ Excel version 16.71 for Mac. Statistical analysis was conducted using IBM SPSS Statistics 29 for Mac. Sample characteristics were summarized using frequency distribution tables and chi-square tests. Descriptive statistics for frequency distribution tables included counts, means, and standard deviations, while chi-square test descriptions were presented in counts and percentages. Independent sample *t*-tests and chi-square tests were utilized to describe between-group respiratory parameters for the NIPPV and NCPAP groups. Within-group changes at admission and 24 h later were assessed using paired sample *t*-tests. The impact of the type of ventilator used and various complications were evaluated, with the NCPAP group serving as the control group. A binary logistic regression model was established, and odds ratios were used to describe the probability of outcome occurrence after adjustment. All tests were two-tailed, and statistical significance was considered when *p*-value was less than 0.05.

## 3. Results

Among a total of 57 premature infants, 3 infants with congenital genetic abnormalities (Beckwith-Wiedemann syndrome, Spinal Muscular Atrophy, and Agenesis of Corpus Callosum) were excluded. The final analysis included 54 premature infants, with 32 in the NCPAP group and 22 in the NIPPV group (Figure 1).

The clinical characteristics of newborns and mothers in the two study groups are presented in Table 1. Newborns in the NIPPV group had smaller gestational ages, lower birth weights, and a higher proportion of females. Additionally, they received MIST earlier after birth, all with *p*-values less than 0.05. The NCPAP group had a higher proportion of newborns delivered by cesarean section and more newborns receiving a second dose of MIST, although these differences did not reach statistical significance. There were no statistically significant differences in the maternal characteristics between NCPAP and NIPPV groups.

The respiratory parameters and severity of RDS of newborns in the two groups before and after receiving MIST are presented in Table 2. There were no significant differences between the two groups in terms of PEEP and FiO_2_ before MIST initiation. Similarly, after MIST, no significant differences were noted. Notably, the NCPAP group showed changes in PIP values after MIST, with 15 infants transitioning to NIPPV and 4 infants requiring endotracheal intubation with invasive positive pressure ventilation (IPPV). In the NIPPV group, 8 infants received endotracheal intubation with IPPV after MIST. Before MIST, a total of 24 newborns using NCPAP and 7 using NIPPV underwent chest radiographic follow-up. The average RDS severity for the NCPAP group was 3.13, while the NIPPV group had an average severity of 2.71. After MIST, a total of 32 newborns using NCPAP and 22 using NIPPV underwent chest radiographic follow-up. The average RDS severity for the NCPAP group was 2.53, and for the NIPPV group, it was 2.50.

The respiratory parameter changes within each group of newborns in the two enrollment groups before and after MIST are presented in Table 3. The results indicate that, in the NCPAP group, the average FiO_2_ decreased from 51.4% before MIST to 34.8% after MIST. In the NIPPV group, the average FiO_2_ decreased from 54.5% before MIST to 34.5% after MIST. Whether in the NCPAP or NIPPV group, newborns tolerated lower oxygen concentrations after MIST, indicating a significant improvement in oxygen saturation for both groups following MIST (*p*-values < 0.05).

The venous blood gas analysis data for newborns in both enrollment groups at admission and 24 h after birth are presented in Table 4. The average blood pH values at 24 h after birth were pH 7.31 for the NCPAP group and pH 7.36 for the NIPPV group (*p*-values < 0.05). Despite the significant difference in pH values, both groups fell within the normal range for newborn blood gas analysis.

The changes in venous blood gas analysis data for newborns in both enrollment groups at admission and 24 h after birth are presented in Table 5. In the NCPAP group, there was a statistically significant increase in pH from 7.22 to 7.31 at admission and 24 h after birth, and a significant decrease in PvCO_2_ from 59.9 mmHg to 45.4 mmHg (*p*-values < 0.05). In the NIPPV group, there was a statistically significant increase in pH from 7.23 to 7.36 and a significant decrease in PvCO_2_ from 56.1 mmHg to 41.1 mmHg at admission and 24 h after birth (*p*-values < 0.05). Regardless of the NCPAP or NIPPV group, the venous blood gas analysis values for newborns improved significantly after admission.

The relevant complications and outcomes of newborns in the two enrollment groups are presented in Table 6. The items include BPD (mild, moderate, and severe), ROP (ROP ≥ stage 3), PDA, PAH, Air leaks, IVH (IVH ≥ grade 3), PVE, PVL, Hydrocephalus, NEC (NEC ≥ stage 2), oxygen and ventilator support on day 28 and at corrected age of 36 weeks, use of xanthine medications, days on ventilator, endotracheal intubation within 72 h, endotracheal intubation within 7 days, ICU stay days, total hospital stay days, mortality, and corrected age at discharge.

In comparison to the NCPAP group, the NIPPV group had a higher proportion (45.4%) of BPD cases, which was higher than that of the NCPAP group (37.5%). However, the NCPAP group had a higher proportion (50%) of severe BPD cases compared to the NIPPV group (40%). Regarding ROP, the NIPPV group had a higher proportion of cases (54.5%) compared to the NCPAP group (25%), and the difference was statistically significant (*p*-value: 0.027). Among those with severe ROP (≥stage 3), the NIPPV group had 5 cases, while the NCPAP group had 1 case, with a significant difference (*p*-value: 0.036). The NIPPV group also showed higher proportions of brain-related complications, PAH, and NEC, but the differences were not statistically significant. Both groups had similar occurrences of PDA.

On day 28 after birth, the NIPPV group had a higher average oxygen concentration (26%, ±10.5%) and a higher proportion of cases requiring ventilator support (68.1%). At the corrected age of 36 weeks, the NIPPV group also had a higher average oxygen concentration (25.8%, ±15.5%) and a higher proportion of cases requiring ventilator support (68.1%). In the NCPAP group, 19 newborns (59.3%) switched to higher support ventilators (15 switched to NIPPV, 4 underwent endotracheal intubation for IPPV), while in the NIPPV group, 8 newborns (36.3%) underwent endotracheal intubation for IPPV. The average total days on ventilator for the NIPPV group were 69.4 days (±63.3 days), higher than the NCPAP group’s average of 37.7 days (±39.2 days), with a significant difference (*p*-value: 0.045). In both groups, 2 newborns each underwent endotracheal intubation within 72 h due to respiratory failure, and no newborns underwent endotracheal intubation between 72 h and 7 days.

In the NCPAP group, 19 newborns (52.8%) required the use of methylxanthines to improve apnea, while in the NIPPV group, 17 newborns (77.3%) required them. However, the statistical difference was not significant.

The average ICU stay days for the NIPPV group were 74.3 days (±57.6 days), compared to the NCPAP group’s average of 41 days (±33.6 days). The average total hospital stay days for the NIPPV group were 86 days (±62 days), longer than the NCPAP group’s average of 54.5 days (±40.2 days), with a statistically significant result (*p*-value: 0.028). Finally, in the NCPAP group, 1 newborn (3.1%) died, while in the NIPPV group, 5 newborns (22.7%) died, with a statistically significant difference (*p*-value: 0.036).

The sample characteristics of the NCPAP group and NIPPV group from Table 1 indicate significant differences in gestational age, birth weight, and gender. Therefore, when establishing statistical models using binary logistic regression for all complications and outcomes, gestational age, birth weight, and gender were used as correction factors. The NCPAP group was set as the control group, and odds ratios for the probability of outcomes were described, as shown in Table 7.

Compared to the NCPAP group, the odds ratios for complications and related outcomes in the NIPPV group did not reach statistical significance. This implies the impact for outcomes has been corrected and adjusted, the chances of complications and other outcomes following MIST were similar in both the NCPAP and NIPPV groups.

## 4. Discussion

According to the American Academy of Pediatrics, for preterm infants at risk of developing neonatal RDS, NCPAP is recommended as the preferred choice for respiratory support. Early administration of exogenous surfactant therapy is also advised [11]. The European Consensus Guidelines, updated in 2022, highly recommend the use of NCPAP or synchronized NIPPV (sNIPPV) as the first choice for respiratory support in neonatal with RDS. If surfactant therapy is required, the guidelines suggest using the LISA technique as the preferred method [12].

In the retrospective medical record study conducted in this research, after assessment by neonatologists, preterm infants received either NCPAP or NIPPV as the primary respiratory system support. Subsequently, exogenous surfactant therapy using the MIST technique was administered based on clinical conditions. Both LISA and MIST are similar administration techniques, aiming to minimize invasive damage to the airways. They involve the use of a flexible and thin tube passed through the vocal cords into the airway, confirmed with a laryngoscope, to deliver surfactant therapy [13]. The study utilized a 16G angiocatheter for the MIST procedure. While there is no conclusive evidence on which type of tube is most successful, some studies indicate that trained neonatologists have a higher success rate in performing this technique [14,15,16]. The physicians performing MIST in this study were on-duty neonatologists who had received training in the technique. However, the experience of each physician varied, potentially influencing the success rate of MIST.

Due to limitations in the number of available ventilators, not all neonates in the NIPPV group could be assigned the same ventilator model. The E360 ventilator provided leak compensation during exhalation with a flow range of 3–8 L/min, while the NPB840 provided leak compensation with a flow range of 1–30 L/min. The inconsistency in ventilator models raises the question of whether it affects the respiratory stability of neonates. However, adjustments to respiratory parameters based on the clinical presentation and physiological assessment were performed by respiratory therapists and neonatologists.

In this study, all enrolled infants received respiratory support using the PSIMV + PS mode, a form of sNIPPV. Corrado et al.’s research indicates that sNIPPV might offer better patient–ventilator synchrony, reduce respiratory effort, and improve respiratory stability compared to other modes [17]. Additionally, animal experiments also suggest that sNIPPV might lead to better outcomes than NCPAP or traditional NIPPV in terms of gas exchange, dynamic compliance, and lung injury [18]. Recent systematic reviews have highlighted that sNIPPV is more effective in reducing the need for reintubation and invasive ventilator support in preterm infants [9,19]. In 2019, European consensus guidelines shifted their recommendation from NCPAP as the first choice to a strong recommendation for either NCPAP or sNIPPV.

The optimal threshold for administering surfactant therapy remains controversial. A recent comprehensive analysis of 58 randomized controlled trials indicates that for ≤30-week preterm infants with RDS requiring FiO_2_ ≥ 40%, surfactant administration may be considered. However, there remains a lack of conclusive evidence to definitively compare the efficacy between FiO_2_ thresholds of 30% and 40% [20]. In Taiwan, due to national health insurance regulations, surfactant therapy is recommended for newborns with RDS only when FiO_2_ is set at 40% or higher, and arterial oxygen pressure cannot be maintained above 80 mmHg, or the oxygen ratio (O_2_ artery/O_2_ alveoli) is less than 0.2. Previous studies suggest that the optimal timing for surfactant therapy is when the FiO_2_ is set below 30% and SpO_2_ cannot be maintained above 90% [16,21,22]. The type of surfactant used can vary, and different brands like Survanta^®^, Curosurf^®^, Infasurf^®^, and Neosurf^®^ contain different concentrations of phospholipids and surfactant proteins. However, only Survanta^®^ is available in Taiwan.

The clinical characteristics of the two groups of enrolled subjects showed statistical differences in gestational age, birth weight, gender, and time of receiving the first dose of MIST after birth (*p* < 0.05) (Table 1). Neonates in the NIPPV group compared to the NCPAP group had smaller gestational age and lower birth weight. Both gestational age and birth weight are risk factors for developing RDS, with smaller gestational age and lower birth weight associated with higher probabilities of RDS. Typically, neonates with smaller gestational age and lower birth weight are more likely to require higher respiratory support, such as NIPPV, upon birth. In this study, neonates in the NIPPV group received the first dose of MIST earlier compared to the NCPAP group, indicating a higher severity of RDS presentations in the NIPPV group, which prompted earlier administration of surfactant therapy.

Comparing the variability of respiratory parameters between the NCPAP and NIPPV groups after implementing MIST, it was found that both groups experienced significant reductions in oxygen concentration after MIST (*p* < 0.001) (Table 2 and Table 3). This indicates that neonates in both groups could maintain oxygen saturation above 90% with lower oxygen concentrations post-MIST, suggesting the efficacy of combining MIST with non-invasive respiratory therapy in improving oxygenation in neonates with RDS.

In this study, the NCPAP group received more doses of surfactant therapy compared to the NIPPV group. Within the NCPAP group, 15.6% of cases required a second dose of surfactant therapy compared to only 9.1% in the NIPPV group. Studies from Turkey and the United States have also shown similar trends, with a higher percentage of NCPAP-treated cases requiring surfactant therapy compared to those treated with NIPPV [23,24]. Compared to NCPAP, NIPPV can provide PIP, allowing for a higher pressure gradient to the distal airway and increasing mean airway pressure. This enables the distal airway to maintain expansion without collapsing during both inspiration and expiration, leading to stable ventilation and oxygenation. This reduces the likelihood of neonates requiring a second dose of surfactant.

In addition, the Cochrane Database of Systematic Reviews compiled six randomized controlled trials comparing the outcomes of early rescue surfactant therapy and late rescue surfactant therapy. Early rescue surfactant therapy was defined as medication administration within 2 h after birth, while late rescue surfactant therapy was defined as medication administration after 2 h of birth. Compared to late rescue surfactant therapy, early rescue surfactant therapy for newborns with RDS requiring ventilatory support can reduce the risk of acute lung injury, as well as decrease neonatal mortality and the risk of chronic lung disease [25]. Indeed, in this study, the timing of the first dose of exogenous surfactant may have contributed to the higher proportion of neonates requiring a second dose in the NCPAP group compared to the NIPPV group. Since the administration of surfactant therapy is based on the regulations, the NCPAP group might have received the first dose of medication later than the NIPPV group, potentially leading to a greater need for subsequent doses in some cases.

Before administering MIST, 24 neonates in the NCPAP group underwent chest radiographic examination, while in the NIPPV group, only 7 neonates underwent the same procedure. Upon admission, some neonates may exhibit severe clinical symptoms prompting our neonatologists to immediately administer surfactant therapy without waiting for a radiographer to conduct portable chest radiography in the intensive care unit. Therefore, not all enrolled neonates were able to undergo chest radiographic examination before MIST to assess the severity of RDS. However, all enrolled neonates underwent chest radiographic examination after MIST administration.

In this study, the number of cases using methylxanthines was higher in the NIPPV group compared to the NCPAP group. This may be attributed to the NIPPV group having neonates with smaller gestational age and lower birth weight compared to the NCPAP group. Due to poorer upper airway muscle tone and incomplete development of central respiratory drive, neonates in the NIPPV group exhibited more severe manifestations of mixed-type apnea. Despite receiving active respiratory support from NIPPV, they still experienced respiratory pauses. Therefore, more neonates in the NIPPV group required methylxanthines to improve respiratory instability.

Additionally, the occurrence rates of complications and related outcomes were higher in the NIPPV group compared to the NCPAP group in this study. Significant differences were observed in the occurrence rate of ROP and ROP ≥ stage 3. The mortality rate was also significantly higher in the NIPPV group. However, logistic regression analysis with adjustments for gestational age, birth weight, and gender did not show statistically significant differences in the occurrence rates of complications and other related outcomes between the two groups.

Furthermore, the NIPPV group had longer durations of respiratory support, ICU stay, and hospital stay compared to the NCPAP group in this study. This may be attributed to the transition from NIPPV to NCPAP as a step towards weaning off respiratory support, potentially prolonging the overall durations of respiratory support, ICU stay, and final hospital stay in the NIPPV group.

This study, in addition to the previously mentioned factors such as different neonatologists performing MIST and the inability to standardize the model of NIPPV ventilators, as well as adjustments of ventilator parameters by either physicians or respiratory therapists, is constrained by several limitations. Firstly, it is a retrospective review of medical records rather than a randomized controlled trial. Neonates were assessed and screened upon admission, with neonatologists selecting the type of ventilator based on clinical presentation. For instance, neonatologists may choose NIPPV for neonates with perceived severe RDS. However, retrospective studies can only collect clinical data under specific conditions, thus making it challenge to accurately observe clinical manifestations at the time of events, such as changes in heart rate before and after MIST. Furthermore, the sample size of this study is insufficient to stratify analysis by gestational age, hindering the comparison of the combined MIST and NIV therapy strategy for RDS between neonates below 28 weeks and those above 28 weeks regarding short-term outcomes. Moreover, not all preterm neonates undergo chest radiography before respiratory therapy, leading to a lack of comprehensive imaging diagnostic results in this study. Additionally, we were unable to gather information on the environmental and genetic factors of the enrolled infants, which are considered influential factors for RDS. The final limitation of our study is the absence of arterial blood gas data for comparison between arterial and venous blood gases under the two conditions, NCAP and NIPPV. In our protocol, upon admission to the neonatal intensive care unit, the standard procedure is to place an umbilical venous catheter for blood sample collection. This methodology led to our exclusive reliance on venous blood gas analyses throughout the study.

## 5. Conclusions

Based on the study findings, it can be concluded that combining MIST with NIV, whether using NIPPV or NCPAP, yielded no significant differences in short-term outcomes for preterm newborns with RDS, including the intubation rate within 72 h, the incidence of complications, and mortality. However, NIPPV as the initial respiratory support resulted in a longer duration of respiratory support and hospital stay. Despite the relatively late administration of exogenous surfactant, opting for a NIPPV PSIMV + PS mode as the ventilation mode appears to be a viable treatment approach for preterm newborns with RDS in Taiwan. Nevertheless, due to the limited number of cases in this study, further investigation with a larger sample size is warranted.

## Figures and Tables

**Figure 1 biomedicines-12-00838-f001:**
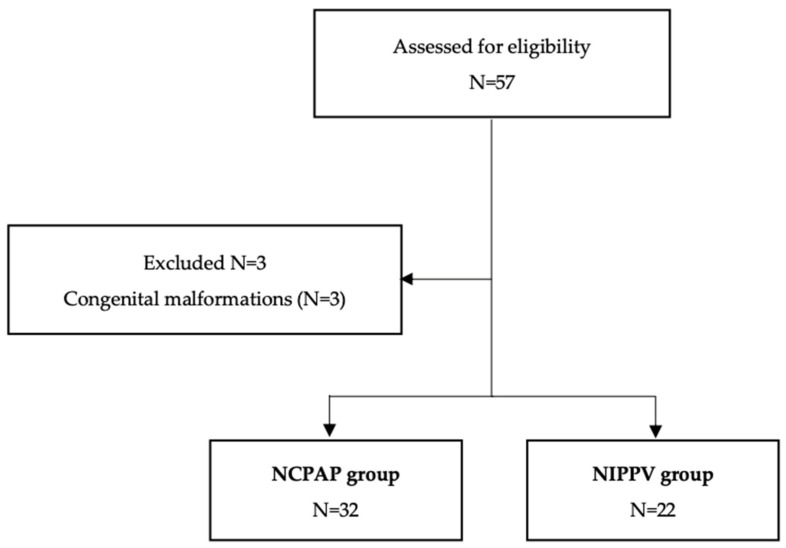
Flowchart of participants.

**Table 1 biomedicines-12-00838-t001:** Patient demographics.

	NCPAP (N = 32)	NIPPV (N = 22)	*p* Value
Preterm infants			
Gestation age (weeks)	30.7 (4.01)	28.3 (3.16)	0.019
Birth weight (g)	1713.6 (729.2)	1206.6 (606.5)	0.010
>1000 g	25 (78.1)	11 (50)	0.031
<1000 g	7 (21.9)	11 (50)	
Male/Female	23/9	9/13	0.023
Singleton	22 (68.6)	13 (59.1)	0.465
Antenatal steroids	22 (68.8)	16 (72.7)	0.753
Antenatal antibiotics	18 (56.3)	16 (72.2)	0.218
Caesarean section	17 (53.1)	8 (36.4)	0.225
Agpar Score 1 min	6.2 (1.5)	4.8 (2.0)	0.389
Agpar Score 5 min	7.7 (1.0)	7.1 (1.4)	0.267
Dose of MIST	1.25 (0.6)	1.09 (0.29)	0.215
1 dose	27 (84.4)	20 (90.9)	0.687
>1 dose	5 (15.6)	2 (9.1)	
Age at first MIST (h)	9.14 (8.6)	2.58 (2.76)	0.001
Mothers			
Age	32.0 (5.3)	33.0 (4.9)	0.511
Taiwanese	31 (96.9)	22 (100)	1.0
Gravida	2.50 (1.6)	2.14 (1.98)	0.882
Para	1.81 (0.93)	1.36 (0.72)	0.102
Abortion	0.69 (1.23)	0.77 (1.47)	0.547

Values are mean (SD), N (%). Student’s *t*-test, Chi-square test. *p* < 0.05.

**Table 2 biomedicines-12-00838-t002:** Respiratory support parameters and severity of RDS before and after MIST.

	Before MIST		*p*	After MIST		*p*
	NCPAP (N = 32)	NIPPV (N = 22)		NCPAP (N = 32)	NIPPV (N = 22)	
PIP (cmH_2_O)	-	14.1 (1.8)	-	13.1 (4.4) *	13.5 (2.8)	0.75
PEEP (cmH_2_O)	4.6 (0.6)	4.3 (0.4)	0.53	4.6 (0.6)	4.4 (0.5)	0.17
FiO_2_ (%)	52.9 (17.3)	55.2 (16.4)	0.63	34.8 (7.5)	34.5 (0.8)	0.89
	NCPAP (N = 24)	NIPPV (N = 7)		NCPAP (N = 32)	NIPPV (N = 22)	
Grade of RDS	3.13 (0.53)	2.7 (0.75)	0.115	2.53 (0.56)	2.50 (0.51)	0.837

* NCPAP to NIPPV: 15, NCPAP to IPPV: 4, NIPPV to IPPV: 8. Values are mean (SD). Student’s *t*-test. *p* < 0.05. NCPAP: nasal continuous positive airway pressure; NIPPV: nasal intermittent positive pressure ventilation; IPPV: invasive positive pressure ventilation; PIP: peak inspiratory pressure; PEEP: positive end-expiratory pressure; FiO_2_: inspired fraction of oxygen.

**Table 3 biomedicines-12-00838-t003:** Variation of respiratory support before and after MIST.

	NCPAP (N = 32)		*p*	NIPPV (N = 22)		*p*
	Before MIST	After MIST		Before MIST	After MIST	
PIP (cmH_2_O)	-	10.5 (7.1)	-	14.1 (1.8) *	13.5 (2.8)	0.07
PEEP (cmH_2_O)	4.6 (0.6)	4.6 (0.6)	1.0	4.3 (0.4)	4.4 (0.5)	0.16
FiO_2_ (%)	51.4 (15.3)	34.8 (7.5)	0.001	54.5 (116.5)	34.5 (9.8)	0.001

* NCPAP to NIPPV: 15, NCPAP to IPPV: 4, NIPPV to IPPV: 8. Values are mean (SD). Paired-*t* test. *p* < 0.05. NCPAP: nasal continuous positive airway pressure; NIPPV: nasal intermittent positive pressure ventilation; IPPV: invasive positive pressure ventilation; PIP: peak inspiratory pressure; PEEP: positive end-expiratory pressure; FiO_2_: inspired fraction of oxygen.

**Table 4 biomedicines-12-00838-t004:** Venous blood gas parameters at admission and at 24 h.

	Blood Gas at Admission		*p*	Blood Gas at 24 h		*p*
	NCPAP (N = 32)	NIPPV (N = 22)		NCPAP (N = 32)	NIPPV (N = 22)	
pH	7.22 (0.05)	7.23 (0.66)	0.46	7.31 (0.07)	7.36 (0.09)	0.02
PvCO_2_	59.9 (13.8)	56.1 (9.9)	0.28	45.4 (8.6)	41.1 (12.1)	0.16
HCO_3_	27.5 (17.4)	23.0 (2.8)	0.24	22.0 (2.5)	22.9 (6.7)	0.49
B.E.	−3.2 (4.8)	−4.5 (3.0)	0.27	−3.7 (2.6)	−3.3 (2.6)	0.55

Values are mean (SD). Student’s *t*-test. *p* < 0.05. NCPAP: nasal continuous positive airway pressure; NIPPV: nasal intermittent positive pressure ventilation; PvCO_2_: venous partial pressure of carbon dioxide; HCO_3_: bicarbonate; B.E.: base excess.

**Table 5 biomedicines-12-00838-t005:** Variations of venous blood gas parameters.

	NCPAP (N = 32)		*p*	NIPPV (N = 22)		*p*
	Admission	24 h		Admission	24 h	
pH	7.22 (0.05)	7.31 (0.07)	0.001	7.23 (0.06)	7.36 (0.09)	0.001
PvCO_2_	59.9 (0.07)	45.4 (2.5)	0.001	56.1 (9.9)	41.4 (12.1)	0.001
HCO_3_	27.5 (17.4)	22.0 (2.5)	0.09	23.0 (2.8)	22.9 (6.7)	0.47
B.E.	−3.2 (4.8)	−3.7 (2.6)	0.60	−4.5 (3.0)	−3.2 (2.6)	0.04

Values are mean (SD). Paired-*t* test. *p* < 0.05. NCPAP: nasal continuous positive airway pressure; NIPPV: nasal intermittent positive pressure ventilation; PvCO_2_: venous partial pressure of carbon dioxide. HCO_3_: bicarbonate; B.E.: base excess.

**Table 6 biomedicines-12-00838-t006:** Outcomes of included participants.

	NCPAP (N = 32)	NIPPV (N = 22)	*p* Value
Comorbidity			
BPD	12 (37.5)	10 (45.4)	0.427
Mild	1 (8)	1 (10)	0.896
Moderate	5 (42)	5 (50)	
Severe	6 (50)	4 (40)	
ROP	8 (25)	12 (54.5)	0.027
ROP (≥stage 3)	1	5	0.036
PDA	24 (75)	17 (77.2)	0.848
PAH	2 (6.2)	4 (18.1)	0.211
Air leaks	2 (6.2)	4 (18.1)	0.211
IVH	6 (18.7)	9 (40.9)	0.074
IVH (≥grade 3)	1	1	1.000
PVE	20 (62.5)	17 (77.2)	0.251
PVL	3 (9.3)	5 (22.7)	0.248
Hydrocephalus	0 (0.0)	2 (9.0)	0.161
NEC	1 (3.1)	2 (9.0)	0.560
NEC (≥stage 2)	0	1	0.407
At 28 days			
FiO_2_ (%)	23.8 (6.6)	26.0 (10.5)	0.360
with support	14 (43.7)	15 (68.1)	0.061
At PMA 36 weeks			
FiO_2_ (%)	22.7 (4.4)	25.8 (15.5)	0.300
with support	17 (53.1)	9 (40.9)	0.397
Require higher respiratory support *	19 (59.3)	8 (36.3)	0.166
Respiratory support days	37.7 (39.2)	69.4 (63.3)	0.045
IPPV days	5.2 (17.4)	19.3 (39.3)	0.126
NIV days	27 (20.9)	37.3 (25.4)	0.108
Intubation within 72 h	2 (6.2)	2 (9.0)	-
Intubation within 7 days	0	0	-
Methylxanthines **	19 (52.8)	17 (77.3)	0.170
ICU days	41.0 (33.6)	74.3 (57.6)	0.210
Hospital stays	54.5 (40.2)	86.0 (62.0)	0.028
Age at discharge	38.6 (4.1)	40.6 (7.8)	0.288
Death	1 (3.1)	5 (22.7)	0.036

* NCPAP to NIPPV: 15, NCPAP to IPPV: 4, NIPPV to IPPV: 8. ** Methylxanthines include caffeine and theophylline. Values are mean (SD), N (%). Student’s *t*-test, Chi-square test. *p* < 0.05. NCPAP: nasal continuous positive airway pressure; NIPPV: nasal intermittent positive pressure ventilation; IPPV: invasive positive pressure ventilation; BPD: bronchopulmonary dysplasia; ROP: retinopathy of prematurity; PDA: patent ductus arteriosus; PAH: pulmonary arterial hypertension; IVH: intraventricular hemorrhage; PVE: periventricular echogenicities; PVL: periventricular leukomalacia; NEC: necrotizing enterocolitis; FiO_2_: inspired fraction of oxygen; PMA: postmenstrual age; IPPV: invasive positive pressure ventilation; NIV: non-invasive ventilation; ICU: intensive care unit.

**Table 7 biomedicines-12-00838-t007:** The outcomes of using NIPPV compared to using NCPAP.

Variables (Reference)	β	S.E.	*p* Value	Odds Ratio	95% C.I. for Exp (β)	
					Lower	Upper
Air leak	1.253	1.039	0.228	3.500	0.456	26.841
BPD	−0.526	0.797	0.509	0.591	0.124	2.818
Moderate to severe BPD	−0.441	0.737	0.549	0.643	0.152	2.727
Respiratory support at day 28	−0.610	1.183	0.606	0.543	0.053	5.527
O_2_ supplement at day 28	−5.741	4.300	0.182	0.003	0.000	14.695
Respiratory support at week 36	−0.872	0.680	0.199	0.418	0.110	1.584
O_2_ supplement at week 36	−0.452	0.823	0.583	0.636	0.127	3.192
Methylxanthines *	−5.096	7089	0.999	0.006	0.000	-
IVH	0.424	0.700	0.544	1.529	0.388	6.024
IVH ≥ grade 3	−4.273	4.326	0.323	0.014	0.000	67.054
PVE	1.101	0.717	0.125	3.008	0.738	12.263
Cystic PVL	1.292	0.980	0.187	3.640	0.534	24.831
Posthemorrhagic hydrocephalus	17.362	5340.852	0.997	34,747,980.8	−0.982	1.006
ROP	1.102	0.886	0.213	3.010	0.531	17.082
ROP ≥ stage 3	2.405	1.369	0.079	11.083	0.757	162.280
NEC	0.619	1.355	0.648	1.858	0.131	26.444
NEC ≥ stage 2	18.139	5499.397	0.997	75,423,990.2	0.000	-
PDA	−1.035	1.007	0.304	0.355	0.049	2.558
Early-onset sepsis	0.638	0.719	0.375	1.892	0.462	7.751
Late-onset sepsis	−1.072	1.077	0.320	0.342	0.041	2.827
PAH	0.468	0.996	0.638	1.597	0.227	11.255
Death	−3.897	2.087	0.062	0.020	0.000	1.214
Higher respiratory support	−1.023	0.663	0.123	0.359	0.098	1.318
More than 30 days of ventilator	0.193	0.821	0.814	1.213	0.243	6.063

* Methylxanthines include caffeine and theophylline. S.E., Standard error of mean. C.I., Confidence Interval. *p* < 0.05. BPD: bronchopulmonary dysplasia; IVH: intraventricular hemorrhage; PVE: periventricular echogenicities; PVL: periventricular leukomalacia; ROP: retinopathy of prematurity; NEC: necrotizing enterocolitis; PDA: patent ductus arteriosus; PAH: pulmonary arterial hypertension.

## Data Availability

The data presented in this study are available on request from the corresponding author due to privacy.

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
