# Peer review of "The Outcomes of Preterm Infants with Neonatal Respiratory Distress Syndrome Treated by Minimally Invasive Surfactant Therapy and Non-Invasive Ventilation"

_biomedicines, 2024, doi:10.3390/biomedicines12040838_

Round 1

Reviewer 1 Report

Comments and Suggestions for Authors

 The authors evaluate the outcomes of Preterm Infants with Neonatal Respiratory Distress Syndrome Treated by Minimally Invasive Surfactant Therapy and Non-Invasive Ventilation. The authors demonstrate that differences were not statistically significant. Analysis of venous blood gas and respiratory parameter changes indicated that both NCPAP and NIPPV groups experienced improvements in oxygenation and ventilation following MIST.

The retrospective experimental design is well constructed and organized also statistically, however I would have evaluated the arterial blood gas rather than the venous one, because it reflects more the oxygenation of the peripheral tissues and evaluates the conditions of hypoxia and shock more significantly.

There are literature data which indicate how RDS can also be influenced by environmental and genetic factors, i.e. there are gene polymorphisms that influence the outcome of the clinical therapeutic course of young patients (Anciuc-Crauciuc M, Cucerea MC, Crauciuc GA , Tripon F, Bănescu CV. Evaluation of the Copy Number Variants and Single-Nucleotide Polymorphisms of ABCA3 in Newborns with Respiratory Distress Syndrome-A Pilot Study. Medicina (Kaunas). 2024 Feb 29;60(3):419. doi: 10.3390 /medicina60030419. PMID: 38541145.)

In addition to the intrinsic factors, the type of treatment could also be considered, for example cortisone during pregnancy and postnatal budesonide together with surfactant.

The conclusions are well proposed but I wonder if the data could be modified by comparing arterial and venous blood gas in the two NCAP and NIPP classes. Furthermore, I would cite the protocol that indicates which FiO2 cut-off values are necessary to decide whether to administer surfactant or not: Ramaswamy VV, Bandyopadhyay T, Abiramalatha T, Pullattayil S AK, Szczapa T, Wright CJ, Roehr CC. Clinical decision thresholds for surfactant administration in preterm infants: a systematic review and network meta-analysis. EClinicalMedicine. 2023 Jul 20;62:102097. doi: 10.1016/j.eclinm.2023.102097. PMID: 37538537; PMCID: PMC10393620.

The references are appropriate

The tables are well done, for the abbreviations it would be appropriate to add a list of the acronyms, with their meaning.

Author Response

Thank you for the insightful suggestions and constructive feedback.

Firstly, we appreciate and concur with your opinion regarding the preference for arterial blood gas analysis over venous. In our study, the standard procedure upon admission to the neonatal intensive care unit involves the placement of an umbilical venous catheter, from which we subsequently draw blood samples. This practice resulted in our study solely utilizing venous blood gas analyses. We have acknowledged this methodological aspect as a limitation within our study and have included a discussion about it in the manuscript (Lines 404-409). For subsequent studies related to this topic, we are certainly open to incorporating the reviewer's recommendation to include arterial blood gas analysis as a measure.

Secondly, we deeply appreciate your insights into both the extrinsic and intrinsic factors influencing RDS. The retrospective design of our study meant that specific data on the environmental and genetic backgrounds of the infants enrolled were not available. We have acknowledged this as a methodological limitation in our study (Lines 402-404). In terms of prenatal cortisone administration, we have included this data in Table 1 under "Antenatal Steroids," noting that there was no significant difference in the use of Antenatal Steroids between the study groups. As for the utilization of postnatal budesonide in conjunction with surfactant therapy, our investigation was constrained by national health insurance policies, which permit surfactant therapy only within the first 48 hours following birth.

Furthermore, we are thankful for the reference provided concerning the threshold for surfactant therapy. Regarding the FiO2 cut-off values, we have incorporated this reference into our manuscript and discussed its significance in the discussion section (Lines 307-311), ensuring that this crucial information is thoroughly addressed.

Reviewer 2 Report

Comments and Suggestions for Authors

Excellent retrospective study on relevant topic, well presented and written manuscript. 

Due to the retrospective nature few novel or insightful conclusions can be drawn. 

Did the neonates in the CPAP group all comply with the legal requirements for surfactant therapy (e. g. FiO2 > 40%? in table 3 the FiO2 in this group is 51.4% with an SD of 15.3, which suggests that some infants did not really meet the criteria)?

Refrence 24 is a Cochrane review of 2000, which has been replaced with a new version in 2012.

I would prefer to see a more detailed paragraph on the limitations of this study in the discussion section. 

Comments on the Quality of English Language

No major concerns. 

Author Response

Thank you for the reviewer's meticulous examination of our statistical results. Upon revisiting the original data, I confirmed that both the NCPAP and NIPPV groups exhibited FiO2 levels exceeding 40% prior to treatment, aligning with the treatment criteria of this study. It's important to clarify that standard deviation is a measure of data dispersion and does not indicate the lower limit of statistical values. Lower limits are more accurately described using different statistical measures, such as minimum values or confidence intervals, as outlined by Wasserman, L. (2004) in "All of Nonparametric Statistics" (Springer-Verlag, ISBN 0-387-40272-1). For further clarity, I have also included the original data items for the reviewer to examine.

 The original data on the FiO2 between the two groups are as follows.

Group(1:NCPAP, 2:NIPPV)

FiO2 before MIST

FiO2 after MIST

1

40

30

1

40

25

1

100

40

1

40

40

1

40

30

1

60

30

1

50

40

1

100

30

1

50

40

1

50

40

1

45

45

1

40

55

1

40

30

1

100

45

1

70

35

1

50

30

1

40

25

1

40

30

1

50

40

1

40

25

1

50

40

1

50

35

1

60

40

1

60

40

1

40

30

1

50

50

1

60

30

1

40

35

1

50

25

1

60

30

1

40

25

1

50

40

2

40

25

2

40

25

2

45

30

2

70

50

2

40

21

2

45

30

2

80

23

2

100

30

2

40

40

2

60

40

2

40

35

2

50

40

2

60

55

2

80

55

2

40

30

2

40

21

2

60

30

2

70

40

2

55

40

2

50

45

2

50

30

2

60

40

I am grateful for the notification about the necessity to update a reference. This has been appropriately addressed and corrected in the manuscript as reference number 25.

Regarding the study's limitations, in addition to the challenges of differing practices among neonatologists performing MIST, the inability to standardize NIPPV ventilator models, and the adjustment of ventilator parameters by physicians or respiratory therapists, this research is a retrospective analysis of medical records rather than a prospective randomized controlled trial. Neonates were evaluated and selected for ventilation type based on clinical assessments at admission, leading to potential selection bias, such as the preference for NIPPV in cases of perceived severe RDS. The retrospective nature limits our ability to capture real-time clinical data accurately, such as heart rate fluctuations before and after MIST application. The sample size of this study is also not large enough to allow for stratified analysis by gestational age, which restricts our ability to compare outcomes of combined MIST and NIV therapy for RDS between neonates below and above 28 weeks gestationally. Furthermore, not all preterm neonates received chest radiography prior to respiratory therapy, resulting in incomplete imaging data. Regarding blood gas analysis, the protocol in our neonatal intensive care unit involves using an umbilical venous catheter for blood sample collection, hence relying exclusively on venous blood gas analyses. This practice resulted in our study solely utilizing venous blood gas analyses. We have incorporated a paragraph into our manuscript to outline and discuss the limitations of our study comprehensively (Lines 388-409).